# The Influence of Maltodextrin on the Thermal Transitions and State Diagrams of Fruit Juice Model Systems

**DOI:** 10.3390/polym12092077

**Published:** 2020-09-12

**Authors:** Pedro García-Coronado, Alma Flores-Ramírez, Alicia Grajales-Lagunes, Cesar Godínez-Hernández, Miguel Abud-Archila, Raúl González-García, Miguel A. Ruiz-Cabrera

**Affiliations:** 1Faculty of Chemical Sciences, Autonomous University of San Luis Potosí, Manuel Nava 6, 78290 San Luis Potosí, Mexico; pgarciam16@gmail.com (P.G.-C.); aalmaramirez@hotmail.com (A.F.-R.); grajales@uaslp.mx (A.G.-L); raulgg@uaslp.mx (R.G.-G.); 2Desert Zones Research Institute, Autonomous University of San Luis Potosí, Altair 200, 78377 San Luis Potosí, Mexico; cesar.godinez@uaslp.mx; 3National Institute of Technology of Mexico, Technological Institute of Tuxtla Gutiérrez, Street Km 1080, Tuxtla Gutiérrez 29050, Mexico; miaba69@hotmail.com

**Keywords:** state diagrams, maltodextrin, thermal transitions, DSC, model systems

## Abstract

The state diagram, which is defined as a stability map of different states and phases of a food as a function of the solid content and temperature, is regarded as fundamental approach in the design and optimization of processes or storage procedures of food in the low-, intermediate-, and high-moisture domains. Therefore, in this study, the effects of maltodextrin addition on the freezing points ($T_{m}^{\prime }$, $T_{m}$) and glass transition temperatures ($T_{g}^{\prime }$, $T_{g}$) required for the construction of state diagrams of fruit juice model systems by using differential scanning calorimetry methods was investigated. A D-optimal experimental design was used to prepare a total of 25 anhydrous model food systems at various dry mass fractions of fructose, glucose, sucrose, pectin, citric acid, and maltodextrin, in which this last component varied between 0 and 0.8. It was found that maltodextrin mass fractions higher than 0.4 are required to induce significant increases of $T_{g}^{\prime }$, $T_{m}^{\prime }$, $T_{g}$, and $T_{m}$ curves. From this perspective, maltodextrin is a good alternative as a cryoprotectant and as a carrier agent in the food industry. Furthermore, solute-composition-based mathematical models were developed to evaluate the influence of the chemical composition on the thermal transitions and to predict the state diagrams of fruit juices at different maltodextrin mass fractions.

## 1. Introduction

A state diagram is a graphical map of the different states of a food or biomaterial as a function of temperature over the entire solid mass fraction scale of materials containing freezable and unfreezable water [1,2,3,4]. In this context, a state diagram usually includes the freezing curve as a function of the solid content ($T_{m}\;\mathrm{vs}.\;w_{s}$), the glass transition curve as a function of the solid content ($T_{g}\;\mathrm{vs}.\;w_{s}$), and the maximal-freeze-concentration condition, defined by the onset melting temperature of ice crystals ($T_{m}^{\prime }$), the glass transition temperature at maximum ice formation conditions ($T_{g}^{\prime }$), and the solid mass fraction ($w_{s}^{\prime }$) [1,2,3,4]. These diagrams have been of great help in monitoring the progress and development of various employed unit operations, such as freezing, frozen storage, lyophilization, cryoconcentration, dehydration, and spray drying, which are all used to extend shelf life and to generate a range of high-, intermediate-, and low-moisture fruit products, such as whole fruits, cut fruits, juices, purees, jams, marmalades, dried fruits, powders, and leathers [1,2,4,5,6,7,8,9]. For instance, the freezing curves ($T_{m}\;\mathrm{vs}.\;w_{s}$) and the freeze-concentrated unfrozen phase transition temperatures $T_{g}^{\prime }$ and $T_{m}^{\prime }$ of products can be used to prevent physical, chemical, and structural changes that take place during the frozen storage of fresh and cut fruits and to avoid the product shrinkage or collapse usually observed during the freeze drying of biological materials [5,10,11,12,13,14]. The temperatures $T_{g}^{\prime }$ and $T_{m}^{\prime }$ are regarded as reference parameters determining the stability of frozen foods, because maximum ice formation takes place when food systems are stored between these temperatures [2,15]. As a general statement, the formation of a glassy vitreous state is then required in frozen storage to prevent molecular motion and further crystallization of water into ice, as well as in freeze drying, because collapse during primary drying will occur when the product temperature exceeds the collapse temperature, which is normally a few degrees above the $T_{g}^{\prime }$ value [10,11,12,13,14]. On the other hand, the relationship between $T_{g}$ and $w_{s}$ in the solid mass fraction domain of $w_{s}^{\prime }\leq T_{g}\leq 1$ has also been regarded in the literature as a reference parameter determining the suitable conditions of drying processes and the storage stability of low-moisture food products. In these cases, the various time-dependent structural transformations, such as the stickiness and deposition events occurring on the dryer surface during spray drying or caking and the crystallization phenomena that take place during fruit powder storage, are highly dependent on $T_{g}$ values [16,17].

For these purposes, several state diagrams of pure components, model food systems, and real fruit products have been reported in the literature [4,18,19,20,21,22,23,24,25,26]. Nevertheless, from the above studies it was found that fruit products presented very low $T_{g}^{\prime }$, $T_{m}^{\prime }$, and anhydrous sample glass transition $\left(T_{gs}\right)$ values, which ranged from −71 to −38 °C, from −52 to −26 °C, and from 12 to 75 °C, respectively. Thus, it is sometimes not possible to design suitable, efficient, and economical processes and frozen storage procedures for sugar-rich products such as fruits [2,11]. The use of polymers as cryoprotectants during frozen storage and as carrier agents in spray drying processes to manipulate the physical state and deliberately elevate the$\;T_{g}^{\prime }$, $T_{m}^{\prime }$, and $T_{g}$ of foods has been widely recommended [11,27,28,29,30,31,32]. Typically, polymers with high molecular weight (HMW), such as maltodextrin, polydextrose, hydrocolloids, and gum, have been used [11,14,28,29,30,31,32,33]. The use of HMW carbohydrates in the construction of state diagrams for fruit products has not been explored extensively, and there are only a few studies in which the state diagrams of some fruits with a maltodextrin addition have been reported [34,35,36]. On the other hand, water and soluble solids such as sugars, pectin, and organic acids are the main fruit components, and the amount of each of these constituents can change drastically from one fruit to another. Furthermore, the number of possible fruit compositions is great. Therefore, experiments with fruit juice model systems with controlled chemical compositions are required. The aim of the present work is (i) to evaluate the effect of maltodextrin on the $T_{g}^{\prime }$, $T_{m}^{\prime }$, $T_{m}$, and $T_{g}$ values of different fruit juice model systems prepared with various water contents by using differential scanning calorimetry (DSC), (ii) to construct the corresponding state diagrams, and (iii) to evaluate the influence of the chemical composition on the abovementioned thermal transitions.

## 2. Materials and Methods

### 2.1. Materials

Analytical-grade maltodextrin dextrose equivalent 4–7 (product No. 419672; molecular weight (MW) 3600, Sigma-Aldrich Co. St. Louis, MO, USA), crystalline fructose (product No. F2543; MW 180.16, Sigma-Aldrich Co. St. Louis, MO, USA), glucose (product No. G7528; MW 180.2, Sigma-Aldrich Co. St. Louis, MO, USA), sucrose (product No. S0389; MW 342.3, Sigma-Aldrich Co. St. Louis, MO, USA), citric acid (product No. 251275; MW 192.12, Sigma-Aldrich Co. St. Louis, MO, USA), and pectin from apples (product No. 76282, MW 208.2, Sigma-Aldrich Co. St. Louis, MO, USA ) were purchased and used without further treatment in the experiments.

### 2.2. Preparation of Fruit Juice Model Systems

A completely randomized D-optimal experimental design for mixtures of six components was used to prepare a total of 25 anhydrous model food systems at various dry mass fractions of fructose ($X_{F}$), glucose ($X_{G}$), sucrose ($X_{S}$), citric acid ($X_{A}$), pectin ($X_{P}$), and maltodextrin ($X_{M}$), as shown in Table 1. The studies carried out by Grajales-Lagunes et al. [4] and Fongin et al. [37,38] were considered as references to establish the mass fractions of citric acid ($0\leq X_{A}\leq 0.15$), pectin ($0\leq X_{P}\leq 0.15$), and maltodextrin ($0\leq X_{M}\leq 0.8$) in the model food systems.

In order to obtain uniform mixing for the component mixtures, the procedure of complete dissolution followed by freeze drying was used for each of the model systems. Thus, solutions with 60% water were prepared, then frozen at −60 °C using a laboratory freezer (ScientTemp Model 86-01A Adrian, Michigan, USA) for 24 h and dried at −40 °C with a 5-mTorr vacuum using a freeze dryer (IlshinBioBase, TFD8501, Seoul, Korea). After complete drying, the freeze-dried mixtures were transferred to a pestle and mortar where they were ground into a fine powder. These samples were then equilibrated over Drierite^®^ (anhydrous calcium sulfate, aw ≈ 0) in desiccators at room temperature for at least 4 weeks to obtain completely dried samples. Afterwards, these equilibrated freeze-dried mixtures were conditioned in the low, intermediate, and high moisture domains following the methodologies proposed by Grajales-Lagunes et al. [4] and Ruiz-Cabrera et al. [22], then subjected to the calorimetric analysis described below, in which the freezing points ($T_{m}^{\prime }$, $T_{m}$) and glass transition temperatures ($T_{g}^{\prime }$, $T_{g}$) for various water contents were determined.

### 2.3. Determination and Modeling of the Thermal Transition by DSC

A Q2000 differential scanning calorimeter (TA Instruments, New Castle, Delaware, USA) equipped with an RCS90 cooling system and Universal Analysis 2000^®^ software for data treatment were used. The measurements were carried out in an inert atmosphere using nitrogen at a flow rate of 50 mL/min. An empty DSC pan as a reference and samples of about 10 mg were used in all of the cases. The standard mode with the linear temperature program was used to determine the freezing points and glass transition temperatures in samples containing freezable water [4,26]. Samples contained in sealed pans were cooled at 20 °C/min from room temperature to −90 °C, equilibrated for 5 min, and then heated to 20 °C at a heating rate of 10 °C/min. For the samples in the concentration range of 20% and 50% water, the previous calorimetry protocol was used with annealing at the apparent value of the freezing point ($T_{m}^{\prime }$−1 °C) as the additional treatment to maximize ice formation in the samples [4,26]. In these cases, the samples were then equalized in the DSC at 20 °C, cooled to −90 °C at 20 °C/min, held for 2 min, warmed at 20 °C/min to the annealing temperature ($T_{m}^{\prime }$−1 °C), kept for 30 min, recooled to −90 °C at 20 °C/min, held for 5 min, and finally scanned to 20 °C at 10 °C/min. The $T_{g}^{\prime }$ and $T_{m}^{\prime }\;$values were assigned to the midpoint of the first and second step changes, respectively, of the observed heat flow and temperature relationship during the heating process, whereas the freezing point ($T_{m}$) was determined from the peak temperature in the melting endotherm [4,22,26]. For anhydrous solids and samples containing unfreezable water, the calorimeter melt-quenching protocol was used [4,22]. The samples contained in sealed pans were first equilibrated in the DSC at 20 °C, then heated at 20 °C/min to the corresponding melting temperature using a holding time of 2 min. Afterwards, the samples were cooled to −90 °C at 20 °C/min and maintained there for 2 min. Finally, the samples were reheated at 10 °C/min again to the corresponding melting temperature. The midpoint $T_{g}$ value was determined using the half-height method from the reheating DSC thermograms.

In order to construct state diagrams for each model system, the Gordon–Taylor (G-T) equation (Equation (1)) and Chen equation (Equation (2)) were used to model the glass transition curve ($T_{g}\;\mathrm{vs}.\;w_{s}$) and freezing point curve ($T_{m}\;\mathrm{vs}.\;w_{s}$), respectively, as follows [2,21,26]:(1)$$T_{gm}=\frac{w_{s}T_{gs}+K\left(1-w_{s}\right)T_{gw}}{w_{s}+K(1-w_{s})}\;$$
(2)$$T_{m}=T_{w}+\left(\frac{\beta }{\lambda_{w}}\right)ln\left(\frac{1-w_{s}-Bw_{s}}{1-w_{s}-Bw_{s}+Ew_{s}}\right)\;$$

In Equation (1), $T_{gm}$, $T_{gs}$, and $T_{gw}$ are the glass transition temperatures of the sample, anhydrous solids, and amorphous water (−135 °C), respectively. In addition, $w_{s}$ is the mass fraction of solids and K is a constant parameter denoting the strength of the interaction between the water and the food solids [1,2]. In Equation (2), $T_{m}$ and $T_{w}$ are the freezing temperatures of the sample and pure water, respectively; β is the molar freezing point constant of water (1860 kg K/kgmol) [2,21,26]. Moreover, $\lambda_{w}$ and $\lambda_{s}$ are the molecular mass values of water and solids, respectively; E is the molecular mass ratio of water-to-solids (λwλs). Finally, B is the ratio of unfreezable water from the total solid content. A nonlinear regression analysis by the least squares method was performed to estimate the parameters $T_{gs}$, K, E, and B by using Microsoft Excel (2016). The goodness of the fitted models was determined by the coefficient of determination (R^2^).

On the other hand, the procedure proposed by Grajales-Lagunes et al. [4] and Zhao et al. [26] was used to estimate the corresponding average values of $T_{g}^{\prime }$ and $T_{m}^{\prime }$, as well as the values of $w_{s}^{\prime }$ from each of the constructed state diagrams.

### 2.4. Statistical Analysis

A polynomial equation (Equation (3)) was used to evaluate the effects of the weight fractions of fructose ($X_{F}$), glucose ($X_{G}$), sucrose ($X_{S}$), citric acid ($X_{A}$), pectin ($X_{P}$) and maltodextrin ($X_{M}$) on the parameters $T_{gs}$, K, E, B, $T_{g}^{\prime }$, $T_{m}^{\prime }$, and $w_{s}^{\prime }$, as follows:(3)$$\begin{array}{ll} y=a_{1}X_{F}+a_{2}X_{G} & +a_{3}X_{S}+a_{4}X_{p}+a_{5}X_{A}+a_{6}X_{M}+a_{7}X_{F}X_{G}+a_{8}X_{F}X_{S}+a_{9}X_{F}X_{P}\\ & +a_{10}X_{F}X_{A}+a_{11}X_{F}X_{M}+a_{12}X_{G}X_{S}+a_{13}X_{G}X_{P}+a_{14}X_{G}X_{A}+a_{15}X_{G}X_{M}\\ & +a_{16}X_{S}X_{P}+a_{17}X_{S}X_{A}+a_{18}X_{S}X_{M}+a_{19}X_{P}X_{A}+a_{20}X_{P}X_{M}+a_{21}X_{A}X_{M} \end{array}$$

Here, y represents the response variables $T_{gs}$, K, E, B, $T_{g}^{\prime }$, $T_{m}^{\prime }$, $w_{s}^{\prime }$. Parameters $a_{1}$−$a_{21}$ are the coefficients from the regression model for the analysis of variance (ANOVA). A confidence level of 95% (*p* < 0.05) with MODDE 7.0 statistical software (Umetrics AB) was used. Equation (3) was reduced to its corresponding pruned forms after the omission of the statistically nonsignificant coefficient values (*p* > 0.10) and mathematical models were developed.

## 3. Results

### 3.1. DSC Thermograms

Figure 1a shows as an example the heating DSC thermograms obtained for the model food system with the chemical composition of 0.45$X_{F}$:0.15$X_{P}$:0.4$X_{M}$ (experiment no. 6, Table 1) and prepared with moisture contents in the range of 27% to 90% wet basis (w.b.).

The thermograms show three thermal events in which the endothermic peak of ice melting was the most visible feature, preceded by two changes in the baseline, both with characteristics typical of a glass transition [11]. It can be observed that the $T_{m}$ value was greatly depressed as the solid content was increased, ranging in this case from 1.4 to −14.2 °C. The melting point is defined as the temperature at which the liquid and solid phases of water at a given pressure are in equilibrium, while the presence of solutes increases the complexity of crystallization and reduces the partial pressure of water. Therefore, the equilibrium between the two phases (ice and water) can only be reached through a reduction in temperature [1,2,39]. It was also evident that the phase change peak becomes smaller as the solid content increases in the sample because of the reduction of the amount of freezable water in the samples. For aqueous sugar solutions, as in this case, it has been suggested that the first transition can be attributed to the glass transition of the maximally freeze-concentrated phase ($T_{g}^{\prime }$), and the second transition can be considered as the beginning of the melting of ice crystals ($T_{m}^{\prime }$) [40]. Figure 1a shows that the first transition was detected between −48.3 and −42.3 °C and the second one between −30.6 and −28.3 °C. Both parameters varied very little when the water content varied between 27% and 90%, as observed by Grajales-Lagunes et al. [4] and Ruiz-Cabrera et al. [22]. Therefore, the corresponding average values of $T_{g}^{\prime }$ and$\;T_{m}^{\prime }$ for each of the samples in Table 1 were estimated, reported, and subjected to statistical analysis using Equation (3), which are discussed later.

Figure 1b also shows as an example the reheating DSC thermograms obtained for the model food system with the chemical composition of 0.45$X_{s}$:0.15$X_{P}$:0.4$X_{M}$ (experiment no. 8, Table 1) and prepared in the reduced moisture range of 0% to 19% w.b. It can be observed that the samples with any moisture content exhibited a clear endothermic shift because of the glass transition, with $T_{g}$ values ranging from −0.3 °C to 75 °C. The $T_{g}$ values became higher as the solid contents were increased in the samples because of the plasticizing effect of water on the glass transition temperature. It can also be seen in Figure 1b that glass transition occurred over a large temperature range (around 40 and 50 °C). These behaviors can be explained by the presence of maltodextrin in the samples. It is assumed that high molecular weight food components such as proteins, starches, and maltodextrin may exhibit glass transition with temperatures as high as 50 °C [2]. It is important to note, however, that good compatibility was achieved for the components, because global $T_{g}$ values were observed in all the samples.

The heating DSC thermograms of the aqueous model systems of exp. nos. 4, 19, 6, 22, and 5 (Table 1) were chosen to illustrate the effects of the maltodextrin concentration on the $T_{g}^{\prime }$, $T_{m}^{\prime }$, and $T_{m}$ values, as shown in Figure 2a. In this case, all the samples were prepared at the same moisture content of 60% w.b., whereas the maltodextrin mass fraction varied at intervals of approximately 0.2 in the concentration range of 0 to 0.8. As expected, the highest $T_{m}$ value (0.8 °C) was obtained for the sample with the highest maltodextrin mass fraction (exp. no. 4), while the lowest $T_{m}$ value (−6.6 °C) was obtained for the sample with no addition of maltodextrin (exp. no. 5). The freezing point depression is directly proportional to the molar concentration of a solution. This indicates that the lower the molecular mass of a solute, the higher the freezing point depression; that is, a high molecular weight substance has fewer molecules per gram than a low molecular weight one, which demonstrates fewer effects on the freezing point ($T_{m}$). Therefore, the depression in the freezing point decreased as the amount of maltodextrin was increased, as shown in Figure 2a. The same trend was observed for the $T_{m}^{\prime }$ values, which varied from −9.4 °C to −38.6 °C. This trend is to be expected, since $T_{m}^{\prime }$ represents the end point of freezing or the beginning of the melting of ice crystals [40]. The values obtained for $T_{g}^{\prime }$, however, which ranged from −46.2 to −54.5 °C, did not highlight a clear tendency as occurred with $T_{m}$ and $T_{m}^{\prime }$ values. In the literature, it has been reported that $T_{m}^{\prime }$ and $T_{g}^{\prime }$ depend strongly on the type and molecular weight of the food components [2,40]. Therefore, this suggests that other factors such as compound interactions are also involved. Note that only one transition at −9.4 °C was observed for the sample of exp. no 4, and the value of $T_{g}^{\prime }$, was not detected in this sample. These findings are similar to those of Flores-Ramírez et al. [11] and Roos and Karel [40], and are tentatively attributed to the overlapping of the glass transition with the ice melting caused by the HMW of maltodextrin.

Because water plays a strong role in the $T_{g}$ of foods and its value is very low (−135 °C), not taking into account its effect, the reheating DSC thermograms of the model systems of exp. nos. 12, 7, 16, 19, and 24 (Table 1) equilibrated over Drierite^®^ were chosen to understand the effects of maltodextrin addition on the glass transition temperatures of anhydrous samples ($T_{gs}$), as shown in Figure 2b. In general, it can be established that all the model systems with added maltodextrin (exp. nos. 12, 7, 16, and 19) exhibited higher $T_{gs}$ values than the model system without maltodextrin (exp. no. 24), which can be attributed to the high $T_{g}$ value of the maltodextrin (170 to 180 °C) [40]. No proportional relationship between $T_{g}$ and the maltodextrin concentration was observed, however, because no significant difference between the $T_{g}$ values was found for the anhydrous samples of exp. no. 12 and exp. no. 7, nor for the samples of exp. no 16 and exp. no. 19 (Table 1). It is well established that the $T_{g}$ values of amorphous materials are mainly affected by the molecular weight, chemical composition, and plasticizer [2,40]. Therefore, it is evident that the plasticizing effects of low molecular weight compounds such as glucose, fructose, and citric acid are also important in fruit juices [2].

### 3.2. State Diagrams

The experimental freezing points and glass transition temperatures measured by DSC were plotted as functions of the solid contents ($w_{s}$) in order to develop the state diagrams of each of the studied samples. As examples, a comparison of the state diagrams obtained for exp. no. 1 (pure fructose), exp. no. 6 (0.45$X_{F}$:0.15$X_{P}$:0.4$X_{M}$), exp. no. 3 (pure sucrose), and exp. no. 8 (0.45$X_{s}$:0.15$X_{P}$:0.4$X_{M}$) are respectively shown in Figure 3a,b.

As expected, a significant increase of the $T_{g}^{\prime }$, $T_{m}^{\prime }$, $T_{m}$, and $T_{g}$ values was observed in the model food systems with added maltodextrin, and such effects can be better appreciated over the entire solid mass fraction scale, as shown in Figure 3. Generally, the predicted $T_{g}$ curves with the G-T equation (Equation (1)) shifted upwards with increased maltodextrin content. In this way, the predicted $T_{gs}$ values for pure fructose and pure sucrose were, respectively, 10.4 °C and 65.4 °C, whereas the corresponding values for the samples of exp. no. 6 and exp. o. 8, both with maltodextrin fractions of 0.4, were 60.3 °C and 107 °C, respectively (Table 2). The $T_{gs}$ values estimated for fructose and sucrose were reasonably consistent with those reported in the literature for both sugars [2,41]. On the other hand, the predicted $T_{m}$ curves with the Chen equation (Equation (2)) were less pronounced for the samples containing maltodextrin (exp. no. 6 and exp. no. 8), which always exhibited higher $T_{m}$ values in the freezable water domain. As previously mentioned, the freezing point depression is highly dependent on the molecular weight of the systems. This reduced freezing point depression may be relevant for industrial processes such as subcooling, in which the partial or total formation of ice crystals is achieved at relatively high temperatures (−4 °C), and thus the availability of water and water activity to slow microorganism growth is reduced. From Figure 3, it is also observed that the added maltodextrin exhibits a greater influence on the $T_{m}^{\prime }$ values than the $T_{g}^{\prime }$ values, because $T_{m}^{\prime }$ represents the end freezing point, which is also highly dependent on the molecular weights of the compounds, as previously discussed. In Figure 3, it can also be verified that both $T_{g}^{\prime }$ and $T_{m}^{\prime }$ exhibited little variation from the solid mass fraction; therefore, average values can be considered in these samples [4,22]. As a general trend, a proportional increase of the maltodextrin concentration was expected for the $w_{s}^{\prime }$ values, because the unfreezable water mass fraction ($w_{w}^{\prime }=1-w_{s}^{\prime }$) should be reduced as the solid content of maltodextrin is increased in the samples. Nevertheless, the opposite compositional dependence of $w_{s}^{\prime }$ has also been established for some samples, such as pure sucrose (Figure 3b, Table 2), which exhibited one of the highest values of $w_{s}^{\prime }$ (0.796 g solid/g sample). Therefore, an experimental method other than the intersection of the average value of $T_{m}^{\prime }$ with the $T_{m}$ curve to accurately determine the maximum freeze concentration is required.

The estimated parameters of the G-T equation (Equation (1)), the Chen equation (Equation (2)), and the maximal-freeze-concentration condition for each of the studied samples are shown in Table 2. From Table 2, it is observed that for the model systems the parameter ranges are as follows:$\;T_{gs}$:10.4 °C–157.3 °C; K: 2.90–10.32; E: 0.0085–0.1103; B: 0.0557–0.297; $T_{g}^{\prime }$: −57.1 °C–−9.5 °C; $T_{m}^{\prime }$:−42.8 °C–−9.5 °C; $w_{s}^{\prime }$: 0.702–0.824 g solid/g sample, with R^2^ values in the range 0.808–0.994. According to the data for several fruits, such as prickly pear cactus, orange, strawberry, pineapple, apple, date fruit, raspberry, and blueberry, performed by Grajales-Lagunes et al. [4], these parameters varied as follows: $T_{gs}$:12.2 °C–74.6 °C; K: 3.02–5.72; E: 0.0178–0.238; B: 0.04–0.1657; $T_{g}^{\prime }$: −58.8 °C–−46.4 °C; $T_{m}^{\prime }$:−50.3 °C–−31.2 °C; $w_{s}^{\prime }$: 0.690–0.847. The upper limits for the $T_{gs}$ and K parameters obtained in this study are well above those reported in the aforementioned study, while the lower limit of the E parameter is lower. Similarly, the upper limits of the $T_{g}^{\prime }$ and $T_{m}^{\prime }$ values are also well above the values reported for model and real systems [4]. These differences can be mainly attributed to the maltodextrin presence in the systems.

The obtained mathematical models, the ANOVA, and the effect of each component on the $T_{gs}$, K, E, B, $T_{g}^{\prime }$, $T_{m}^{\prime }$, and $w_{s}^{\prime }$ values are given in Table 3. Equation (3) was used to interpret data variability with the determination coefficients, standard deviation, and coefficient of variation, respectively, in the ranges of 0.833–0.999, 0.00682–6.22, and 1.22–23.2. In addition, significance values varied between 0.2143 and <0.0001 (Table 3). From Table 3, it is observed that $T_{gs}$, K, E, B, $T_{g}^{\prime }$, $T_{m}^{\prime }$, and $w_{s}^{\prime }$ were all linearly affected by the mass fractions of the components, followed by binary interactions, with the parameters $T_{g}^{\prime }$ and $w_{s}^{\prime }$ being the most affected. Taking into account the regression coefficients of the models, it can be seen that pectin ($X_{P}$) and citric acid ($X_{A}$) are the most important variables affecting all parameters, while the interactions of maltodextrin ($X_{M}$) with the other components play important roles in the $T_{m}^{\prime }$ values, and the interactions of maltodextrin and pectin with other components have important effects on the parameter B. Regarding the ANOVA, it can be observed (Table 3) that all of the models are significant (P (F > F_0_)), with the exception of the parameter $w_{s}^{\prime },\;$and there is a high probability that other factors not included in the model (noise) affect this response variable. The high values obtained for R^2^ indicated that good control in the performance of the experiments and in the parameter determinations was achieved. On the other hand, the coefficients of variation shown in Table 3 generally indicated that there is homogeneity among the obtained data.

As an example, the performance of the empirical equations given in Table 3 can be observed in Figure 4, where a comparison between the predicted state diagrams at different maltodextrin mass fractions for the model food system 0.283$X_{F}$:0.283$X_{G}$:0.283$X_{S}$:0.075$X_{A}$:0.075$X_{P}$ (experiment no.5, Table 1) was performed. The solid composition of experiment no. 5. was chosen because sucrose, fructose, glucose, pectin, and citric acid are the main solutes of fruit and vegetables, and the maltodextrin mass fraction in the sample was varied between 0 and 0.6. In this case, it was found that the higher the value of the maltodextrin mass fraction $(X_{M}$), the higher the values of $T_{g}^{\prime }$, $T_{m}^{\prime }$, and $w_{s}^{\prime }$, and the less pronounced the curvature of the $T_{m}$ curve. On the other hand, although the curvatures of the predicted $T_{g}$ curves of samples containing maltodextrin were more pronounced compared with the sample without maltodextrin, no significant differences between the $T_{g}$ curves were found when maltodextrin mass fractions of 0.3 and 0.6 were used. A dominant role of the $T_{g}$ of amorphous mango pulp at a higher maltodextrin weight fraction than 0.7 was also observed by Fongin et al. [38]. Perhaps this is one of the reasons why high concentrations of maltodextrins are used as carrier agents in spray drying processes. In this context, the solute-composition-based mathematical models resulting from this study are relevant for predicting state diagrams to monitor the progress and development of various processes, such as freezing, refrigeration, and drying.

## 4. Conclusions

The effects of maltodextrin addition on $T_{g}^{\prime }$, $T_{m}^{\prime }$, $T_{m}$, and $T_{g}$ the during the construction of state diagrams of several fruit juice model systems were investigated. Increasing the maltodextrin mass fraction resulted in a significant increase of the abovementioned thermal transitions. Maltodextrin mass fractions higher than 0.4, however, are required to induce a significant increase of $T_{g}^{\prime },$
$T_{m}^{\prime }$, $T_{m}$, and $T_{g}$ curves. Maltodextrin, therefore, can be considered as a good alternative in the formulation of cryoprotective media for adequate frozen preservation of high- and intermediate-moisture foodstuffs and as a carrier agent in the spray drying process. The developed mathematical models facilitated the determination of the influence of the chemical composition on the $T_{gs}$, K, E, B, $T_{g}^{\prime }$, $T_{m}^{\prime }$, and $w_{s}^{\prime }$ values, and could also be used to predict the state diagrams of samples as a function of the concentrations of solutes predominant in fruit juices and the maltodextrin weight fraction. In this context, the solute-composition-based mathematical models resulting from this study are relevant to the design and optimization of processes and storage procedures for fruit products in the low-, intermediate-, and high-moisture domains.

## Figures and Tables

**Figure 1 polymers-12-02077-f001:**
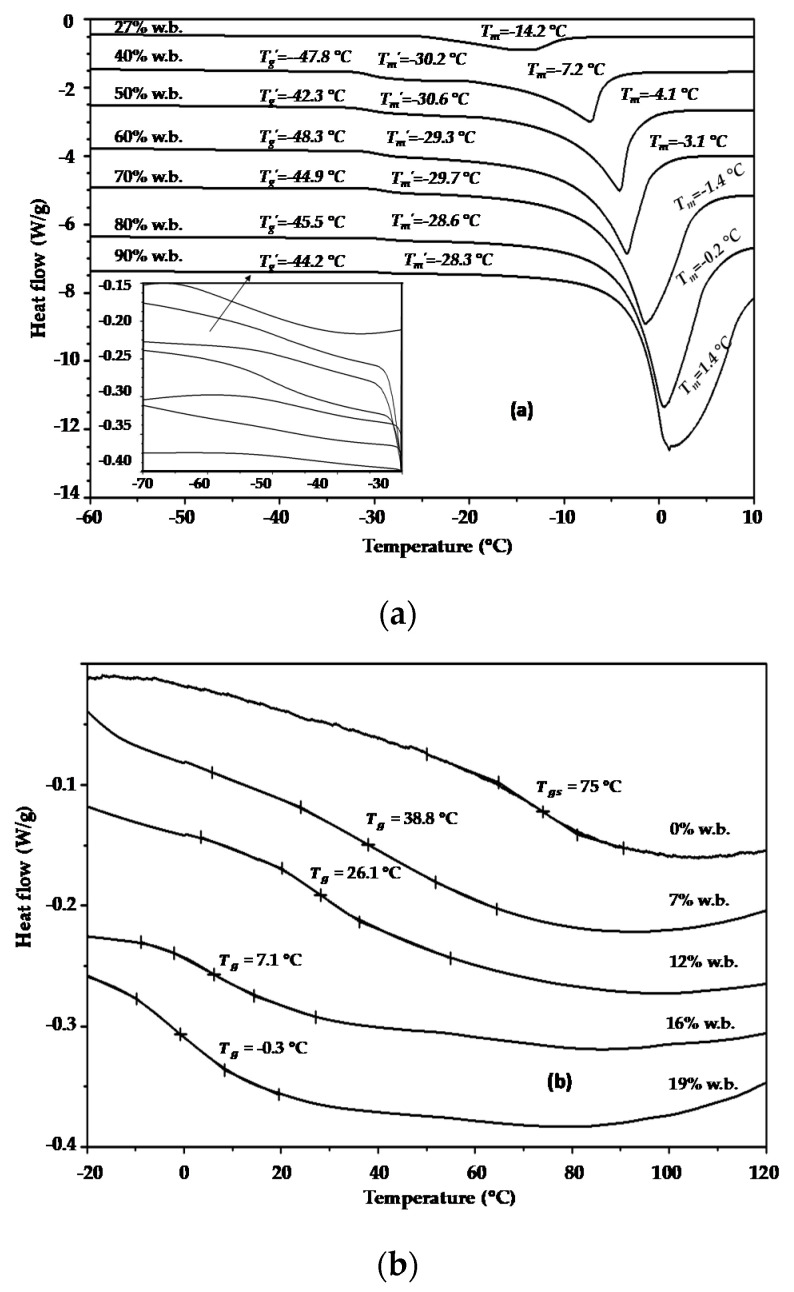
Differential scanning calorimetry (DSC) thermograms used for thermal analysis in model food systems: (**a**) the determination of $T_{g}^{\prime }$, $T_{m}^{\prime }$, and $T_{m}$ in experiment no. 6 (0.45$X_{F}$:0.15$X_{P}$:0.4$X_{M}$) containing freezable water; (**b**) the determination of $T_{g}$ in experiment no. 8 (0.45$X_{s}$:0.15$X_{P}$:0.4$X_{M}$) containing unfreezable water.

**Figure 2 polymers-12-02077-f002:**
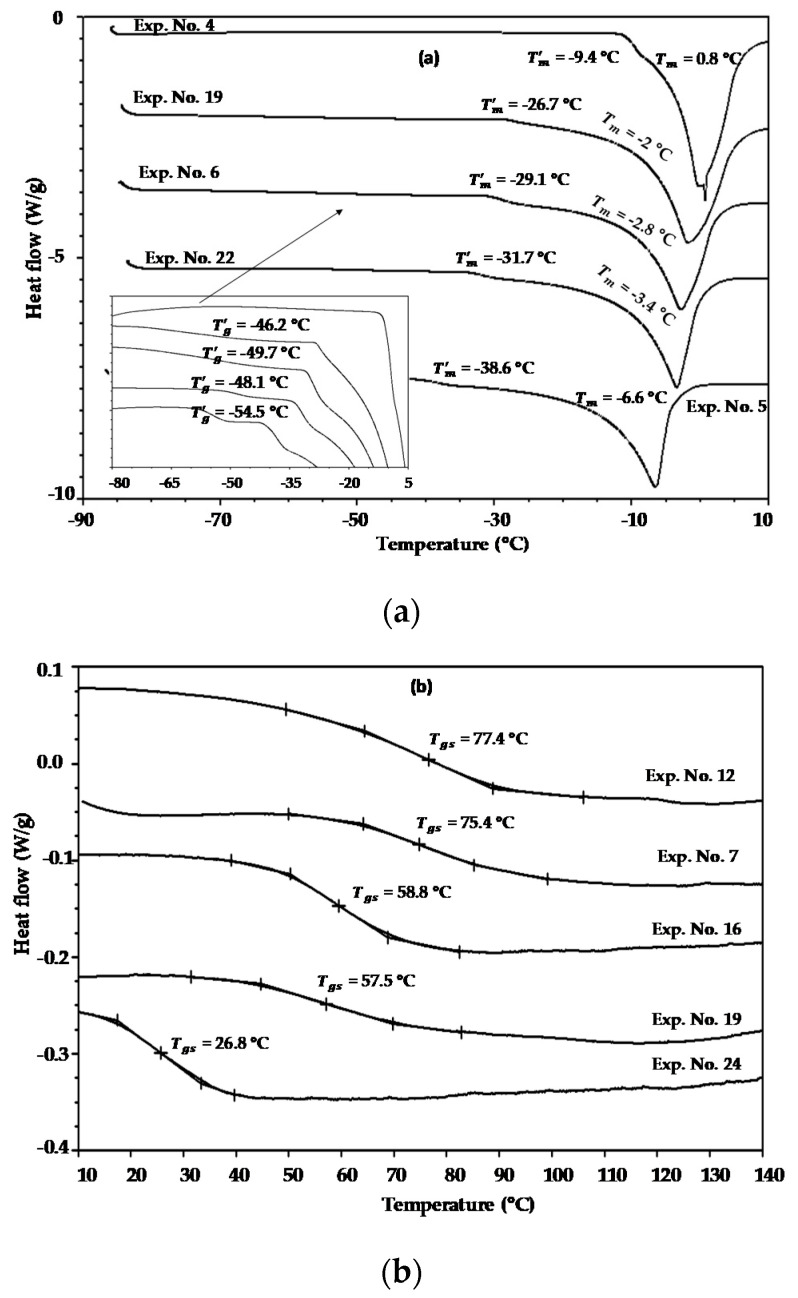
DSC thermograms used for thermal analysis in model food systems: (**a**) the effects of maltodextrin concentration on the $T_{g}^{\prime }$, $T_{m}^{\prime },$ and $T_{m}$ values of samples prepared at 60% w.b.; (**b**) the effects of maltodextrin concentration on the $T_{gs}$ values of anhydrous samples.

**Figure 3 polymers-12-02077-f003:**
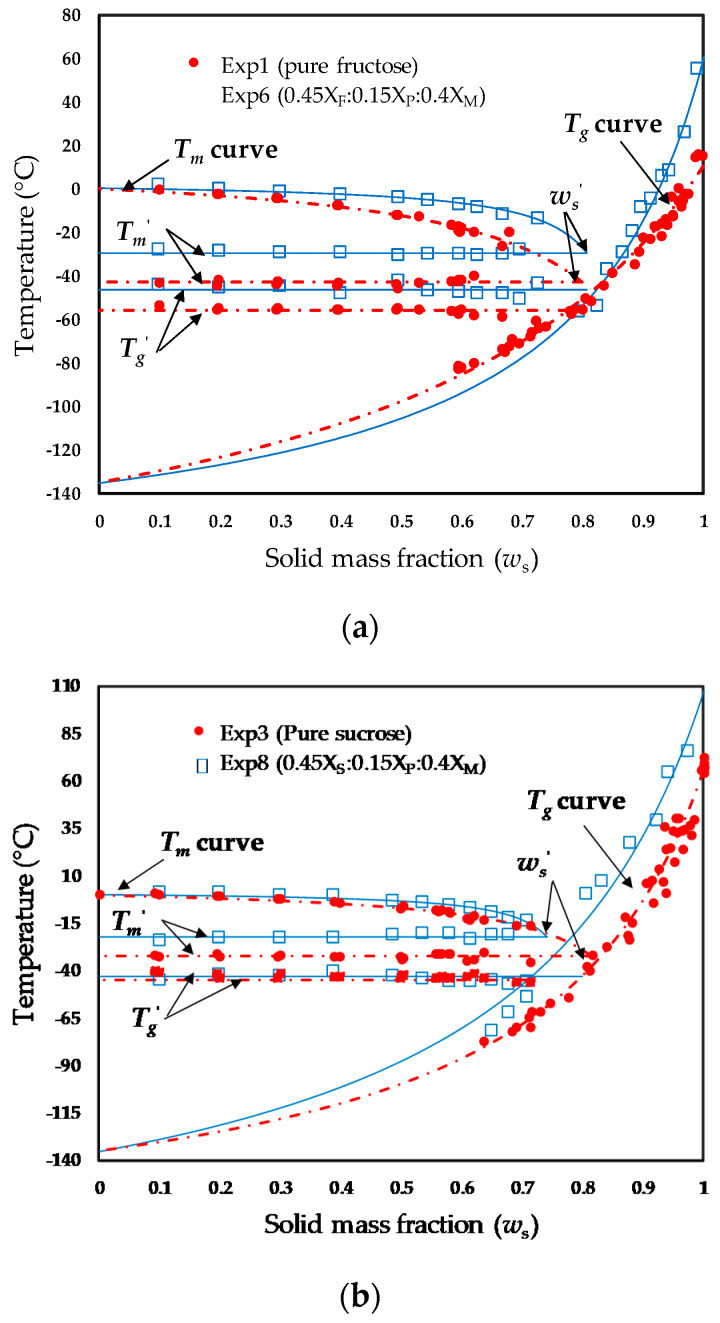
Examples of state diagrams obtained for fruit juice model systems: (**a**) comparison of experiment no. 1 (pure fructose) and experiment no. 6 (0.45$X_{F}$:0.15$X_{P}$:0.4$X_{M}$); (**b**) comparison of experiment no. 3 (pure sucrose) and experiment no. 8 (0.45$X_{s}$:0.15$X_{P}$:0.4$X_{M}$).

**Figure 4 polymers-12-02077-f004:**
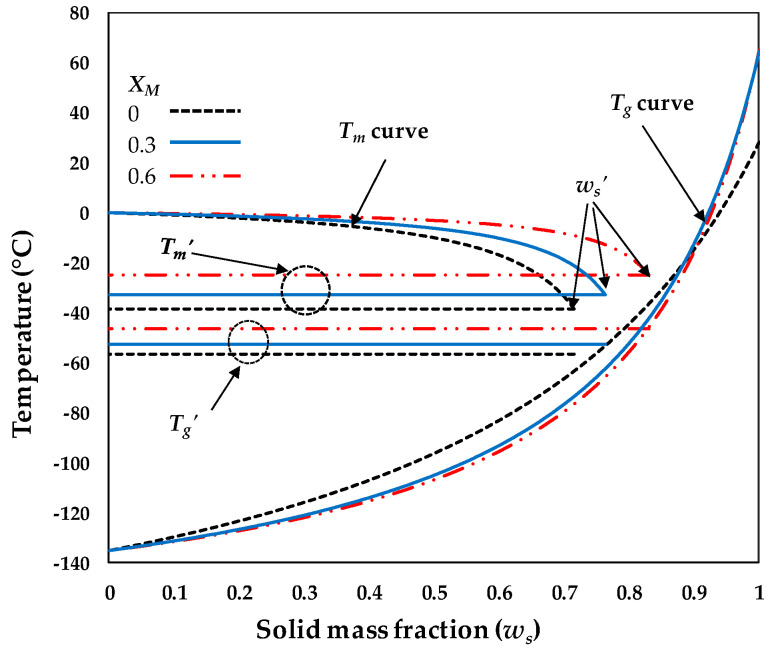
Influence of the maltodextrin mass fraction on the predicted state diagram for the fruit juice model system of experiment no. 5 using $X_{M}$ = 0, 0.3 and 0.6.

**Table 1 polymers-12-02077-t001:** D-optimal experimental design for a mixture of six components and the elaboration of anhydrous model food systems.

Experiment		Mass Fractions					
no.	Ro	$X_{F}$	$X_{G}$	$X_{S}$	$X_{P}$	$X_{A}$	$X_{M}$
1	15	1	0	0	0	0	0
2	6	0	1	0	0	0	0
3	8	0	0	1	0	0	0
4	4	0	0	0	0.15	0.05	0.80
5	1	0.283	0.283	0.283	0.075	0.075	0
6	7	0.45	0	0	0.15	0	0.40
7	21	0	0.45	0	0.15	0	0.40
8	2	0	0	0.45	0.15	0	0.40
9	23	0.50	0	0.50	0	0	0
10	22	0	0.50	0.50	0	0	0
11	18	0.50	0.50	0	0	0	0
12	24	0.145	0.145	0.145	0.059	0.15	0.355
13	20	0.70	0	0	0.15	0.15	0
14	5	0	0.70	0	0.15	0.15	0
15	11	0	0	0.70	0.15	0.15	0
16	25	0.327	0.327	0.077	0.035	0.035	0.198
17	19	0.327	0.077	0.327	0.035	0.035	0.198
18	12	0.077	0.327	0.327	0.035	0.035	0.198
19	3	0.177	0.077	0.077	0.035	0.035	0.598
20	9	0.502	0.077	0.077	0.035	0.110	0.198
21	13	0.077	0.540	0.077	0.035	0.073	0.198
22	14	0.077	0.077	0.577	0.035	0.035	0.198
23	10	0.35	0.35	0	0.15	0.15	0
24	17	0.35	0	0.35	0.15	0.15	0
25	16	0	0.35	0.35	0.15	0.15	0

no = number, Ro = Run order.

**Table 2 polymers-12-02077-t002:** Fitting parameters of Equations (1) and (2), and parameters of the maximal-freeze-concentration condition (MFCC).

Experiment	Parameters of Equation (1)			Parameters of Equation (2)			MFCC		
No.	$T_{gs}$ (°C)	K	R^2^	E	B	R^2^	$T_{g}^{\prime }$ (°C)	$T_{m}^{\prime }$ (°C)	$w_{s}^{\prime }$
1	10.4	2.90	0.986	0.0954	0.1668	0.991	−55.9	−42.8	0.739
2	31.8	3.79	0.994	0.1103	0.0557	0.968	−55.7	−41.8	0.783
3	65.2	4.68	0.975	0.0584	0.0980	0.964	−43.1	−32.5	0.796
4	157.3	10.32	0.808	0.0085	0.1595	0.929	−9.5	−9.5	0.801
5	25.6	3.33	0.990	0.0781	0.2178	0.991	−56.4	−38.7	0.716
6	60.3	5.59	0.983	0.0371	0.1280	0.971	−46.6	−29.6	0.807
7	61.6	5.75	0.964	0.0535	0.0741	0.974	−54.8	−30.2	0.812
8	107	4.09	0.970	0.0256	0.2442	0.975	−44.6	−22.3	0.741
9	24.7	3.53	0.981	0.0877	0.0697	0.950	−50.2	−38.4	0.791
10	41	3.85	0.984	0.0786	0.1076	0.971	−52.0	−38.8	0.781
11	19.1	3.52	0.976	0.0956	0.0936	0.972	−54.6	−42.6	0.781
12	64	6.12	0.876	0.0517	0.1399	0.969	−52.0	−32.0	0.780
13	16.4	3.73	0.954	0.0734	0.1958	0.987	−56.9	−39.3	0.738
14	34.4	4.04	0.987	0.0806	0.1805	0.987	−57.1	−39.0	0.737
15	46.6	4.16	0.951	0.0673	0.1340	0.974	−50.4	−34.7	0.768
16	37.7	4.18	0.899	0.0706	0.1770	0.983	−55.9	−36.9	0.746
17	66	5.44	0.904	0.0552	0.1953	0.966	−51.6	−34.8	0.750
18	73.5	6.41	0.890	0.0553	0.2107	0.988	−52.3	−34.4	0.740
19	47.6	5.06	0.884	0.0385	0.0770	0.967	−47.4	−25.5	0.824
20	52.4	4.43	0.963	0.0529	0.2970	0.986	−55.5	−36.1	0.702
21	67.1	6.09	0.886	0.0700	0.1644	0.985	−55.4	−36.5	0.752
22	92.4	6.90	0.971	0.0408	0.2569	0.981	−48.9	−31.6	0.730
23	18.8	3.75	0.949	0.0841	0.1467	0.985	−56.4	−38.6	0.751
24	36.6	3.43	0.976	0.0583	0.2462	0.984	−53.5	−35.1	0.719
25	54.2	4.78	0.965	0.0801	0.0865	0.977	−55.8	−35.3	0.780

**Table 3 polymers-12-02077-t003:** Solute composition-based mathematical models and ANOVA (*p* < 0.05).

Model	ANOVA			
	P (F > F_0_)	R^2^	S. D.	C.V. (%)
$T_{gs}=8.31X_{F}+29.98X_{G}+58.68X_{S}+1295.36X_{P}-885.31X_{A}+10.52X_{M}\;\;\;\;\;\;\;\;\;\;\;-2139.92X_{F}X_{P}+1653.18X_{F}X_{A}-2555.95X_{G}X_{P}\;\;\;\;\;\;\;\;\;\;\;+2109.04X_{G}X_{A}+122.3X_{G}X_{M}-3002.49X_{S}X_{P}\;\;\;\;\;\;\;\;\;\;\;+2526.09X_{S}X_{A}+468.11X_{S}X_{M}$	<0.0001	0.983	6.22	11.86
$K=2.9X_{F}+3.76X_{G}+4.48X_{S}-130.1X_{P}+119.4X_{A}+3.59X_{M}+109.55X_{F}X_{P}\;\;\;\;\;\;\;\;\;\;\;-106.31X_{F}X_{A}+72.36X_{G}X_{P}-67.19X_{G}X_{A}+12.55X_{G}X_{M}\;\;\;\;\;\;\;\;\;\;\;+28.65X_{S}X_{M}+119.8X_{P}X_{A}+249.97X_{P}X_{M}-248.14X_{A}X_{M}$	<0.0001	0.984	0.31	6.42
$E=0.946X_{F}+0.1019X_{G}+0.0617X_{S}+0.0543X_{P}+0.0648X_{A}-0.0003\;\;\;\;\;\;\;\;\;\;\;-0.1471X_{F}X_{P}-0.0719X_{S}X_{M}$	<0.0001	0.94	0.00682	10.61
$B=0.1413X_{F}+0.0645X_{G}+0.0956X_{S}-9.0225X_{P}+3.9767X_{A}-0.0164X_{M}\;\;\;\;\;\;\;\;\;\;\;+8.3373X_{F}X_{P}+8.0965X_{G}X_{P}+7.8839X_{S}X_{P}\;\;\;\;\;\;\;\;\;\;\;+1.0128X_{S}X_{M}+14.3016X_{P}X_{M}-9.8217X_{A}X_{M}$	0.0018	0.833	0.04	23.25
$T_{g}^{\prime }=-55.95X_{F}-55.56X_{G}-43.16X_{S}-2063.09X_{P}+1103.46X_{A}-34.36X_{M}\;\;\;\;\;\;\;\;\;\;\;+4.43X_{F}X_{G}-1.41X_{F}X_{S}+2027.37X_{F}X_{P}-1249.55X_{F}X_{A}\;\;\;\;\;\;\;\;\;\;\;-23.35X_{F}X_{M}-11.19X_{G}X_{S}+1899.63X_{G}X_{P}\;\;\;\;\;\;\;\;\;\;\;-1130.07X_{G}X_{A}-22.25X_{G}X_{M}+1921.26X_{S}X_{P}\;\;\;\;\;\;\;\;\;\;\;-1169.78X_{S}X_{A}-4.93X_{S}X_{M}+1987.27X_{P}X_{A}\;\;\;\;\;\;\;\;\;\;\;+2819.84X_{P}X_{M}-2024.59X_{A}X_{M}$	<0.0001	0.999	0.61	1.2
$T_{m}^{\prime }=-42.92X_{F}-42.31X_{G}-33.39X_{S}-72.78X_{P}+13.8X_{A}-10.8X_{M}\;\;\;\;\;\;\;\;\;\;\;-17.99X_{F}X_{M}-21.09X_{G}X_{M}-15.8X_{S}X_{A}+134.61X_{P}X_{M}\;\;\;\;\;\;\;\;\;\;\;-172.4X_{A}X_{M}$	<0.0001	0.994	0.69	2.02
$w_{s}^{\prime }=0.75X_{F}+0.78X_{G}+0.80X_{S}+51.35X_{P}-36.73X_{A}+0.72X_{M}+0.08X_{F}X_{G}\;\;\;\;\;\;\;\;\;\;\;+0.03X_{F}X_{S}-53.72X_{F}X_{P}+39.01X_{F}X_{A}+0.05X_{F}X_{M}\;\;\;\;\;\;\;\;\;\;\;+0.02X_{G}X_{S}-52.92X_{G}X_{P}+38.22X_{G}X_{A}-0.30X_{G}X_{M}\;\;\;\;\;\;\;\;\;\;\;-53.85X_{S}X_{P}+39.16X_{S}X_{A}-0.40X_{S}X_{M}-19.73X_{P}X_{A}\;\;\;\;\;\;\;\;\;\;\;-65.01X_{P}X_{M}+57.76X_{A}X_{M}$	0.2143	0.921	0.02	2.94

P (F > F_0_) = Fisher probability; S.D. = Standard deviation; C.V. = Coefficient of variation.

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
