# Peer review of "The Influence of Maltodextrin on the Thermal Transitions and State Diagrams of Fruit Juice Model Systems"

_polymers, 2020, doi:10.3390/polym12092077_

Round 1
Reviewer 1 Report
Dear Authors
In my opinion the aim of study is interesting. It is well realized. Results are well described and interpreted. Conclusions are clear and results from experimental dates.
Below short comments:
Authors include too many References; for example: line 70 is 6 positions, in line 44 is 8 positions, line 48: 6 positions. I propose changes this.
In line 144 is: “1860 kg oC/mol” Please inform about source of this. The unit of this isn`t SI system! Temperature unit is K no oC.
In scientific paper should be this same number of decimal places. Then in line 194 should be: “from -0.28 oC to 74.90 oC” This same in lines 253, 278, or in Tab.2 No 8.
In Tab.3 (model 3) and line 293 should be 0.00682.
Author Response
All the responses are given in the attached document

Reviewer 2 Report
The paper "The influence of maltodextrin on the thermal transitions and state diagrams of fruit juice model systems" by Garcia-Coronado et al. deals with the establishment of a state diagram based of a model system.
The paper is well written, clear and the conclusions are supported by the results. However, minor corrections are needed before its publication:
- the term Xs from eq 1 and 2 may be confunsed with the mass fraction of fructose...maybe the authors may find another adnotation
- line 202: DSC instead of DCS
- line 308: maltodextrin instead of maltodetrin
- line 316: I think that there are no dot between maltodextrin and no...
- it will be very interesting if the authors can propose a general state diagram and not only specific examples (see fig 3)
In view of the above, I recommend the publication of this manuscript after minor corrections.
Author Response

(The authors gave the same response as above.)
